# Faces of Pain during Dental Procedures: Reliability of Scoring Facial Expressions in Print Art

**DOI:** 10.3390/brainsci11091207

**Published:** 2021-09-14

**Authors:** Frank Lobbezoo, Xuan Mai Lam, Savannah de la Mar, Liza J. M. van de Rijt, Miriam Kunz, Maurits K. A. van Selms

**Affiliations:** 1Department of Orofacial Pain and Dysfunction, Academic Centre for Dentistry Amsterdam (ACTA), University of Amsterdam and Vrije Universiteit Amsterdam, 1081 LA Amsterdam, The Netherlands; f.lobbezoo@acta.nl (F.L.); x.m.lam@student.vu.nl (X.M.L.); s.c.u.dela.mar@student.acta.nl (S.d.l.M.); l.j.m.vande.rijt@acta.nl (L.J.M.v.d.R.); 2Department of Medical Psychology and Sociology, Universität Augsburg (UNIA), 86135 Augsburg, Germany; miriam.kunz@med.uni-augsburg.de

**Keywords:** orofacial pain, non-verbal pain expressions, dental print art, reliability

## Abstract

Background: Observational tools have been developed to assess pain in cognitively impaired individuals. It is not known, however, whether these tools are universal enough so that even pain depicted in print art can be assessed reliably. Therefore, the aim of this study was to assess the reliability in scoring facial expressions of pain in dental print art from the 17th, 18th, and 19th century, using a Short Form of the 15-item Pain Assessment in Impaired Cognition (PAIC15-SF) tool. Methods: Seventeen prints of patients undergoing dental procedures were scored twice by two inexperienced observers and an expert and once by a Gold Standard observer. Results: All observers achieved high intra-observer reliability for all four items of the category “facial expressions” and for three items of the category “body movements” (ICC: 0.748–0.991). The remaining two items of the category “body movements”, viz., “rubbing” and “restlessness”, were excluded from further research because it was not possible to calculate a reliable ICC. Overall, the intra-observer reliability of the expert was higher than that of the inexperienced observers. The inter-observer reliability scores varied from poor to excellent (ICC: 0.000–0.970). In comparison to the Gold Standard, the inter-observer reliability of the expert was higher than that of the inexperienced observers. Conclusion: The PAIC15-SF tool is universal enough even to allow reliable assessment of facial expressions of pain depicted in dental print art.

## 1. Introduction

Pain, and especially orofacial pain, is a common condition among the general population [1]. Orofacial pain can originate from dentoalveolar tissues, the masticatory muscles, the temporomandibular joints, and nerve tissues [2]. In most cases, the presence of pain will be addressed verbally in order to receive a proper diagnosis and cure. However, for those persons who suffer from pain but are at the same time limited in their communicative abilities, pain can become undetected. For example, when a patient is cognitively impaired or has dementia, the assessment of pain becomes increasingly challenging due to the loss of communication abilities [3,4]. In such cases, healthcare providers have to rely on observations and intuition to assess when more or different care is needed [5]. In order to support healthcare providers, several observational pain measurement tools, such as the Orofacial Pain Scale for Non-Verbal Individuals (OPS-NVI) [6,7] and the Pain Assessment in Impaired Cognition tool (PAIC15) [8] were developed to identify pain in cognitively impaired individuals. The PAIC15 is composed of three widely accepted categories of non-verbal pain responses, namely five items on facial expressions, five items on body movements, and five items on vocalisations. Research has shown good psychometric properties of the PAIC15; hence, the tool can be considered reliable, valid, and ready to be used in everyday clinical settings [8].

For the periods before photography and cinema, graphic art was used to visualise the daily life of citizens of all classes. Especially the Flemish and Dutch painters of the 17th, 18th, and 19th centuries painted a great number of medical and dental scenes. Among them, symptoms of tooth disease (e.g., a swollen cheek) and the activities of dentists treating them have been used as themes in many works of art. Apart from the fact that the painter was able to transmit valuable information regarding the working conditions of the dentist, displaying dentistry as a mere trade [9], the facial expressions of patients with toothache undergoing dental treatment were painted meticulously. The detailed study of those paintings, drawings, and engravings can therefore provide us with valuable information about non-verbal pain expressions.

Since it is as yet unknown whether observational pain measurement tools such as the OPS-NVI and the PAIC15 are universal enough to assess pain in various groups of patients and in different cultures reliably, we put this question to test as to see whether, even in print art, these tools yield reliable outcomes. Thus, the aim of this study was to assess the intra-observer and inter-observer reliability of two inexperienced observers (viz., dental students), one experienced observer (viz., a dentist with specific expertise in the non-verbal aspects of orofacial pain), and one Gold Standard observer (viz., a world-leading expert in pain-related non-verbal behaviour) in scoring pain-related facial expressions and body movements in dental print art of the 17th, 18th, and 19th century, using an abbreviated version (‘Short Form’) of the PAIC15 (i.e., the PAIC15-SF). Acceptable values for reliability, and especially the inter-observer reliability as compared to the Gold Standard, are needed to conclude that the PAIC15-SF is universal enough as to even allow reliable assessment of facial expressions of orofacial pain depicted in dental print art.

## 2. Material and Methods

### 2.1. Study Design

The study sample consisted of 17 prints depicting dental procedures from the 17th, 18th, and 19th centuries, selected by a world-leading expert in pain-related non-verbal behaviour, viz., Prof. Dr. M. Kunz. The prints were obtained from two almost identical books on dental print art from the 1470–1870 period, composed by Dr. G.J. Schade [10,11]. The Dutch edition, published in 2014, contains 276 prints divided into 100 dental scenes [10]. In 2020, an English version of the same book was published, this time containing 256 prints [11]. For this study, only the most realistic dental scenes of procedures performed by dental and oral surgeons, in which the face of the patient was meticulously painted, were selected by the expert; non-realistic prints (e.g., caricatures and torture scenes) were not selected. From the English edition, the following 15 prints were included in the study: 27.2 (p. 89), 39.3 (p. 167), 41.1 (p. 124), 42.2 (p. 128), 44.1 (p. 135), 49.2 (p. 158), 53.2 (p. 168), 57.2 (p. 234), 69.1 (p. 215), 73.1 (p. 225), 77.1 (p. 237), 77.4 (p. 240), 80.2 (p. 247), 82.2 (p. 251), 88.1 (p. 265), 88.2 (p. 265), and 95.1 (p. 279); from the Dutch edition, the prints 39.3 (p. 167) and 57.2 (p. 234) were selected (these were not included in the English version). Eight prints were full colour, and nine were black and white. The prints’ dimensions varied between 10.5 × 13.5 cm for the smallest one and 36 × 30 cm for the largest one. Three additional prints were used for training purposes: 34.2 (p. 105), 40.2 (p. 123), and 60.2 (p. 185; all from English edition). The Institutional Review Board of the Academic Centre for Dentistry Amsterdam (ACTA), Amsterdam, The Netherlands, granted approval for this study (protocol number 2020186).

### 2.2. Measurement Instrument

The facial expressions as displayed on the selected prints were assessed for possibly expressing pain by using the Dutch version of the PAIC15. This instrument was translated into Dutch from the source language (viz., English) and was culturally adapted to the Dutch situation, using the forward–backward approach of the Guidelines for Establishing Cultural Equivalence of Instruments [5,12], by the Department of Public Health and Primary Care, University Medical Centre, Leiden, The Netherlands. Due to the fact that this study focused on print art, scoring vocalisations was not possible. Thus, the category “vocalisations” was omitted from the tool. Furthermore, all items related to opening the mouth as an expression of pain (category “facial expression”) were also omitted because most prints concerned dental procedures during which the mouth was already opened. Consequently, an abbreviated version (‘Short Form’) of the PAIC15 was used, viz., the PAIC15-SF. The remaining nine items that were used in this study are the facial expressions (4 items) and body movements (5 items), as shown in Table 1. All items of the PAIC15-SF are assessed using an ordinal scale that distinguishes the following grades: 0 = not at all, 1 = slight degree, 2 = moderate degree, 3 = great degree, and NS = not scoreable.

### 2.3. Training

Two observers, both of them third-year dental students at ACTA, were provided training in order to evaluate whether or not they were capable of accurately assessing facial expressions of pain. Their trainer was an experienced user of the PAIC15, viz., Dr. L.J.M. van de Rijt, a dentist with specific expertise in the non-verbal aspects of orofacial pain. As a first step, the students were asked to watch the PAIC15 e-lecture [13]. The e-lecture starts with a short online introduction about the PAIC15, after which the e-training part starts. During this training part, the three categories to score non-verbal pain are clarified by videos, in which actors mimic realistic situations. First, the five facial expressions of pain (viz., frowning, narrowing of the eyes, raising the upper lip, the opening of the mouth, and looking tense) are introduced and clarified. Second, the five different categories of body movements are discussed (viz., freezing, guarding, resisting care, rubbing, and restlessness). Finally, vocalisation in association with pain and its categories (viz., the use of pain-related words or exclamations such as shouting, groaning, mumbling, and complaining) are explained.

During the training, the students familiarised themselves with the PAIC15 by evaluating several training videos, followed by a test. The test consisted of ten movies, in which the students scored all categories using the PAIC15. Two weeks later, the students evaluated the videos again and completed the test for a second time. If the students reached the 70% mark or higher as compared to the gold-standard scores of the videos, they were considered competent enough to evaluate signs that might suggest pain in non-verbal individuals. Both students reached that mark on both occasions, with scores of 84.5% and 87.5%, and 89.3% and 87.5% for the first and second tests, respectively.

### 2.4. Reliability Assessment

By using the PAIC15-SF, the 17 prints were evaluated by the two dental students (observer 1 and observer 2). Despite their thorough training (see above), they were considered inexperienced. The two students were instructed to score all prints independently during the first assessment, followed by an interval of two weeks of no scoring. After this interval, the two students completed a second assessment. During the interval, they did not have access to the data to blind them for their own scores as well as for each other’s scores. Additionally, an experienced observer (expert), Dr. L.J.M. van de Rijt, scored all prints two times on two occasions as well, also using the PAIC15-SF. Finally, the scores of a world-leading expert in pain-related non-verbal behaviour (Prof. Dr. M. Kunz) were used (Gold Standard). Of all scientists in the world, she has published most extensively on this topic (viz., 51 papers, found on 16 July 2021, in Web of Science (WoS) with the query “Dementia AND pain AND facial”). The Gold Standard scored all prints on one occasion.

The collected data were analysed as follows. First, the intra-observer reliability for both inexperienced observers and the expert was calculated. Second, the first evaluation assessments were used to calculate the inter-observer reliability of both inexperienced observers and the expert in comparison to each other, as well as in comparison to the Gold Standard:

Observer 1 versus observer 2;

Observer 1 versus the expert;

Observer 1 versus the Gold Standard;

Observer 2 versus the expert;

Observer 2 versus the Gold Standard;

The expert versus the Gold Standard.

### 2.5. Statistical Analysis

The intra-observer and inter-observer reliability were analysed by means of Intraclass Correlation Coefficients (ICCs), using a two-way mixed model with absolute agreement. ICCs < 0.4 were considered as poor, ICCs between 0.4 and 0.75 were defined as fair-to-good, and ICCs > 0.75 were considered excellent [14]. A confidence interval with a level of 95% was used for the calculations. SPSS version 26 Software (IBM Corp., Armonk, NY, USA, 2019) was used for all statistical analyses, and the significance level was set at *p* < 0.05.

## 3. Results

### 3.1. Intra-Observer Reliability

In Table 1, the intra-observer reliability scores between the first and second assessments are shown as determined individually for the inexperienced observers 1 (Obs. 1) and 2 (Obs. 2) as well as for the experienced observer (Expert). All test-retest reliability scores, quantified as ICC, were determined for the nine different items of the PAIC15-SF. It can be gathered from this table that both inexperienced observers showed a fair-to-good to excellent intra-observer reliability (ICC 0.748–0.991) for all items, except for the body movements “rubbing” and “restlessness”. These latter items were scored too infrequently to calculate a reliable ICC. In addition, it can be concluded from Table 1 that the expert was slightly more stable in the scoring of pain between the two sessions (higher ICCs for most items).

### 3.2. Inter-Observer Reliability

In Table 2, the inter-observer reliability scores are shown for “observer 1 in comparison to observer 2”, “observer 1 in comparison to the expert”, “observer 1 in comparison to the Gold Standard”, “observer 2 in comparison to the expert”, “observer 2 in comparison to the Gold Standard”, and “the expert in comparison to the Gold Standard”. For all comparisons, the scores of the first assessment were used because the intra-observer reliability was already proven to be at least fair-to-good (see above). Again, the items “rubbing” and “restlessness” were scored too infrequently to calculate a reliable ICC.

It can be gathered from Table 2, first data column, that the ICCs for the comparison between observer 1 and observer 2 can be qualified as fair-to-good to excellent (ICC 0.448–0.940). The second column in Table 2 depicts the ICCs for the comparison between observer 1 and the expert; the third column those for the comparison between observer 2 and the expert. All ICCs for these comparisons can be qualified as fair-to-good to excellent (ICC 0.416–0.886 and ICC 0.452–0.884, respectively). The comparison between the expert and the Gold Standard (Table 2, 6th column) resulted in overall higher ICCs for all items in comparison to inter-observer reliability scores of the Gold Standard with observer 1 (Table 2, 4th column) and observer 2 (Table 2, 5th column).

Looking at the agreements between “observer 1 vs. Gold Standard” and “observer 2 vs. Gold Standard”, it appeared that both inexperienced observers had comparable ICCs: in columns 4 (Obs. 1 vs. Gold Standard) and 5 (Obs. 2 vs. Gold Standard), the ICCs scores are all fair-to-good or excellent (ICC 0.722–0.933 and ICC 0.708–0.957, respectively), except for the item “freezing” (ICC 0.347 and ICC 0.000, respectively). In comparison to “observer 1 vs. Gold Standard” and “observer 2 vs. Gold Standard”, the ICC scores for the comparison between the expert and the Gold Standard showed overall higher values (0.630–0.970) for all items, including the item “freezing”.

## 4. Discussion

The aim of this study was to assess whether observational pain measurement tools such as the PAIC15-SF are universal enough even to allow reliable assessment of facial expressions of pain depicted in print art. This was determined by assessing the intra-observer and inter-observer reliability of several observers with varying degrees of experience in scoring pain-related facial expressions and body movements in dental print art of the 17th, 18th, and 19th centuries. To that end, two inexperienced observers, one experienced observer (expert), and one Gold Standard observer evaluated 17 prints included in a book on dental print art [10,11]. The results show that the intra-observer reliability of both inexperienced observers and of the experienced observer (expert) were all excellent, except for the item “raising the upper-lip” scored by observer 1, which had fair-to-good reliability. Due to the fact that the expert’s ICC scores for the inter-observer reliability in comparison with those of the Gold Standard (expert vs. Gold Standard) were overall higher than those of the inexperienced observers (Obs. 1 vs. Gold Standard; Obs. 2 vs. Gold Standard), it can be assumed that the expert is more reliable in establishing facial expressions of orofacial pain in print artwork compared to the inexperienced observers.

The results show that there was an overall high agreement in inter-observer reliability for both inexperienced observers compared to the expert and the Gold Standard. However, the items “freezing”, “rubbing”, and “restlessness” from the category “body movements” either scored lower (freezing) or could not be assessed at all (rubbing and restlessness) for the inter-observer reliability. A possible explanation might be that all these items are difficult to score in print art due to the fact that print art is motionless (i.e., static), whereas these items represent a dynamic activity. This may explain why the ICCs scores for the inter-observer reliability measurements between the observers for scoring “facial expressions” was higher than for scoring “body movements”.

Not a lot of research has been performed on the assessment of orofacial pain in non-verbal individuals [3,4], and no research on this topic has been carried out concerning (print) art. However, several studies have discussed the PAIC-15 and other tools regarding their ability to assess the presence of orofacial pain in non-verbal individuals reliably. For example, the study carried out by de Vries et al. included 237 video clips of Dutch nursing home residents suffering from dementia during their meals [15]. The Orofacial Pain Scale for Non-Verbal Individuals (OPS-NVI) [6,7] was used to assess the potential presence and intensity of orofacial pain experienced by the residents while eating. De Vries et al. (2016) found a good intra- and inter-observer reliability for the “chewing” subscale [15]. However, obtaining the video recordings and interpreting them for pain-related facial expressions in the abundant presence of all kinds of non-pain-related facial expressions was challenging for the researchers. In another study carried out by Delwel et al. [6], the OPS-NVI was used to assess the experienced orofacial pain by patients with dementia during live observations. Unfortunately, that study showed an overall low degree of orofacial pain, most likely because a substantial proportion of the patients were using pain medication. Clearly, taken the limitations of these two previous works into consideration, determining the reliability of the observations of pain-related behaviour was more challenging in those previous studies than in the present one. Nevertheless, the overall results of most previous studies show good reliability for all categories. In the study of Kunz et al. [8], 33 PAIC items were reviewed and showed overall good inter-observer reliability, although for the category “facial expressions”, the inter-observer reliability was lower compared to the categories “body movements” and “vocalisations”. Furthermore, the study by de Waal et al. [16] showed that the reliability for the categories “body movements” and “vocalisations” were higher than several items from the category “facial expressions”. Both outcomes [8,16] are remarkable because in the present study, the items from the category “facial expressions” scored higher inter-observer reliability than the category “body movements”. A possible explanation for this discrepancy might be that behavioural items such as those in the category “body movements” are easier and better to score when assessing a dynamic activity rather than when assessing static print art.

### 4.1. Strengths and Limitations

The first and most important strength of this study is that this is the first study in which print art is being used for assessing the reliability of interpreting expressions of orofacial pain. Second, the PAIC15-SF is extracted from the PAIC15. The PAIC15 was derived from widely recognised observational scales that were all used to identify pain (including orofacial pain) in non-verbal individuals [8,16], which makes the PAIC15-SF the most suitable tool to achieve the purpose of this study. However, it should be noted that the PAIC15 has so far only been tested in European countries [8]. In other words, it is unknown if this observational pain assessment scale can assess pain in individuals from different, non-European cultures. Third, the study consisted of prints depicting patients undergoing dental procedures whilst possibly showing orofacial pain and excluding other habitual facial expressions, which might not be related to pain as was possibly the case in the study by de Vries et al. (2016) [15]. Fourth, the study sample in the present study offered the possibility to assess the intra-observer reliability, just like studies using video clips (e.g., de Vries et al., 2016) [15].

The first limitation of this study might be that the sample of this study was small. There were only 17 prints selected from the books on dental print art [10,11]. Evaluation of the reliability of inexperienced observers in a larger sample size and from different sources of print art is recommended for further studies. Second, a choice could be made to assess not only print art but also different art forms, such as sculptures. Third, the fact that artists may have felt free to modify the actual expressions that they observed while producing the artwork should be taken into consideration when observations such as the ones in the present study are being interpreted.

### 4.2. Implications

This study found that expressed pain as depicted in print art can be assessed reliably by using an abbreviated version of the PAIC15, which emphasises the fact that the PAIC15-SF tool is universal enough to allow a reliable assessment of pain-indicative behaviour even when using depictions of orofacial pain in print art. In addition, it shows that print art can be very true to life. This knowledge may be used by companies to assess the quality of art or by art historians to assess the level of pain humans have been experiencing throughout history. It may also enrichen one’s life because this tool may enlighten the way in which art can be experienced. It can also be derived from this study that it is possible to use print art to partly train inexperienced dental students. It is not always feasible to teach the students how to assess and recognise orofacial pain experienced by non-verbal patients during the clinical training. Training by using print art displaying faces of pain might be an addition to the already existing curriculum.

## 5. Conclusions

The Pain Assessment in Impaired Cognition tool (PAIC15) is developed to assess the presence and degree of pain in patients that suffer from impaired cognition. In this study, a Short Form of the PAIC15 (i.e., the PAIC15-SF) tool was used to assess the presence of orofacial pain experienced by patients depicted in print art. All observers achieved high intra-observer reliability for all four items of the category “facial expressions”, and for three items of the category “body movements”. The remaining two items of the category “body movements”, viz., “rubbing” and “restlessness”, were excluded from further research because it was not possible to calculate a reliable ICC. Overall, the intra-observer reliability of the expert was higher than that of the inexperienced observers. The inter-observer reliability scores varied from poor (for the item “freezing”) to excellent. In comparison to the Gold Standard, the expert scored overall higher ICCs than the inexperienced observers. Hence, the PAIC15-SF tool is universal enough even to allow reliable assessment of facial expressions of pain depicted in dental print art.

## Figures and Tables

**Table 1 brainsci-11-01207-t001:** Intraclass correlation coefficient (ICC) scores for the intra-observer reliability measurements of two inexperienced observers and one experienced observer in scoring pain-related facial expressions and body movements.

	Obs. 1	Obs. 2	Expert
**Facial expressions**			
Frowning	0.907	0.915	0.970
Narrowing of the eyes	0.866	0.924	0.955
Raising the upper lip	0.748	0.771	0.908
Tense impression	0.857	0.914	0.875
**Body movements**			
Freezing	0.833	0.845	0.875
Guarding	0.933	0.988	0.933
Resisting care	0.887	0.991	0.899
Rubbing	*	*	*
Restlessness	*	*	*

Used items of the PAIC15-SF (4× facial expressions, 5× body movements). “Obs.”: Observer. * Excluded from statistical analysis because this item was scored too infrequently.

**Table 2 brainsci-11-01207-t002:** Intraclass correlation coefficients (ICCs) scores for the inter-observer reliability measurements between the observers in scoring pain-related facial expressions and body movements. Scores are based on the first assessment.

	Obs. 1 vs. Obs. 2	Obs. 1 vs. Expert	Obs. 2 vs. Expert	Obs. 1 vs. Gold Standard	Obs. 2 vs. Gold Standard	Expert vs. Gold Standard
**Facial expressions**						
Frowning	0.657	0.886	0.827	0.853	0.855	0.970
Narrowing of the eyes	0.940	0.861	0.858	0.795	0.873	0.943
Raising the upper lip	0.604	0.650	0.884	0.789	0.718	0.856
Tense impression	0.766	0.656	0.722	0.782	0.836	0.884
**Body movements**						
Freezing	0.448	0.416	0.452	0.347	0.000	0.630
Guarding	0.895	0.651	0.805	0.933	0.957	0.811
Resisting care	0.596	0.703	0.754	0.722	0.708	0.779
Rubbing	*	*	*	*	*	*
Restlessness	*	*	*	*	*	*

“Obs.”: Observer. ***** Excluded from statistical analysis because this item was mentioned too infrequently.

## Data Availability

The data presented in this study are available on request from the corresponding author.

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
