# Peer review of "Faces of Pain during Dental Procedures: Reliability of Scoring Facial Expressions in Print Art"

_brainsci, 2021, doi:10.3390/brainsci11091207_

Round 1
Reviewer 1 Report
The Ms brainsci-1335256 entitled “Faces of pain during dental procedures: reliability of scoring facial expressions in print art” is an original research paper reporting on the universality of the Short Form of the 15-item Pain Assessment in Impaired Cognition (PAIC15-SF) tool in the assessment of facial expressions of pain depicted in dental print art. In fact, vocalizations, as well as items concerned with opening the mouth, have been omitted since the assessment has been performed on print art. The study aims at evaluating intra-observer and inter-observer reliability of this tool through two inexperienced observers and one experienced observer in comparison with a world-leading expert Gold Standard observer.
Therefore, the novelty of the Ms is clear. The study has demonstrated the universality of this observational pain assessment tool in scoring as to reliably orofacial pain indicators even in print art.
The title and the keywords reflect the content of the study reported in the Ms.
In the introduction the Authors clearly illustrate the concept of under detection of orofacial pain in non-communicative patients in view of the lack of self-reporting. This is a fundamental issue being very common in general population and very much so in cognitively impaired patients, e.g. people suffering from dementia. Therefore, according to the opinion of this referee, the Ms is relevant to research in this field. Furthermore, a tool able to detect orofacial pain in non-communicative patients, endowed with good validity and reliability and feasible for clinical use is a fundamental device in the armamentarium of pain assessment in this fragile population. The identification of non-verbal pain expressions in print art is a very interesting and innovative approach to underscore the universality of a pain assessment tool.
The methods are described in detail and very accurate, including the PAIC15-SF forward-backward translation and cultural adaptation of the Guidelines for Establishing Cultural Equivalence of Instruments. The use of the obtained PAIC15-SF is very clear. The reporting of the e-training with actors miming the situations is useful for researchers to train observers. The Authors demonstrate and report the possibility to use print art with which to train students and this represents an added value. The blinding of the observers is a very important aspect making rigorous the methodology and the achievement of the results. Also, statistical analysis is correct. I commend the Authors for the novel approach and for the methods adopted.
The results demonstrate fair-to-good to excellent intra-observer reliability for the inexperienced observers and higher ICCs for most items for the expert. They have led to exclusion of “rubbing” and “restlessness” items due to lack of a reliable ICC, having been not frequent to assess. Despite the overall high agreement, for inter-observer reliability the agreement with the Gold Standard has resulted higher for the expert than for the two inexperienced observers; the latter observation underscores the importance of experience for observational assessment of pain.
In the Discussion paragraph, the critical and careful analysis of similar studies previously reported in the literature in conjunction with the declared limitations of the study are appreciated.
Minor points:
Typo at line 294 Conclusions: …presence degree…
Author Response
We sincerely thank the reviewer for the time and attention spent in carefully reading our manuscript. We do hope that our paper adds attention to the identification of non-verbal pain expressions by means of a pain assessment tool.
Reviewer 2 Report
Dear Authors
Thank you for sharing your interesting study. It is highly important for healthcare providers in all domains in an aging society to be educated and trained in pain recognition in people who can not provide verbal and or cognitive expression of pain. So, it is clearly an important study.
Yet there are some points that need further clarification
Material & Methods:
Study design:
- Inexperienced observers: 3rd year students are supposed to already be educated in recognizing non- verbal pain expression as they already treat people. I would rather use less experienced.
- Why did the world leading expert choose those specific 17 prints and not the other 83? And why only 17?
Line 59 &Conclusions:
PAIC15-SF was tested on European countries. your study tested the instrument on Flemish Dutch paintings by Dutch students. In order to call it universal it should be tested on different cultures (e.g., North, Central and south America, Africa, Asia etc. )
Discussion and implication:
Assessment of pain expression in non- verbal cognitive impaired person should be multi facete/item dynamic evaluation using all 3 items as recommended by PAIC -15. Relying only on facial expression and the obvious limitation of using only certain body movements gives limited and maybe not specific information on dental pain expression, as narrowing of the eyes, frowning might not be pain specific. doi: 10.1002/ejp.1536. Epub 2020 Feb 11. PMID: 31981281 and doi: 10.2147/CIA.S144651). Also, by using printed art you have one frame shot of a moment the painter choose to zoom on leaving the “looking tense” to a subjective single point interpretation of the painter. How can you assess the degree/intensity of pain relying on that?
Author Response
We thank the reviewer for the careful reading and interpreting of our manuscript and the constructive remarks. We have taken the comments on board to improve and clarify the manuscript. Please find below a detailed point-by-point response to all comments.
Material & Methods:
Study design:
- Inexperienced observers: 3rd year students are supposed to already be educated in recognizing non- verbal pain expression as they already treat people. I would rather use less experienced.
Even though the reviewer is right about this aspect, we prefer not to change the terminology for the reason that both students had missed the opportunity of treating many patients suffering from oral and orofacial pain due to the Covid pandemic. Thereby, treatment of patients with these complaints is not part of the bachelor curriculum at ACTA. Rather, such patients are being seen during the master phase of the study. - Why did the world leading expert choose those specific 17 prints and not the other 83? And why only 17?
Indeed, only 17 prints were selected by the expert. The main reason is that we aimed to focus only on the most realistic dental scenes of procedures performed by dental and oral surgeons, in which, on top of that, the face of the patient was meticulously painted. Unfortunately, many prints included in the (Dutch/ English) book did not fulfil these criteria. For example, many prints were caricatures, others depicted scenes of torture, or scenes from a play in a theatre. We added more information in order to better clarify this aspect.
Line 59 & Conclusions:
PAIC15-SF was tested on European countries. your study tested the instrument on Flemish Dutch paintings by Dutch students. In order to call it universal it should be tested on different cultures (e.g., North, Central and south America, Africa, Asia etc. )
The reviewer is right about the fact that the PAIC15 has so far only been tested in European countries. We added information to clarify the need for testing in different cultures.
Discussion and implication:
Assessment of pain expression in non- verbal cognitive impaired person should be multi facete/item dynamic evaluation using all 3 items as recommended by PAIC -15. Relying only on facial expression and the obvious limitation of using only certain body movements gives limited and maybe not specific information on dental pain expression, as narrowing of the eyes, frowning might not be pain specific. doi: 10.1002/ejp.1536. Epub 2020 Feb 11. PMID: 31981281 and doi: 10.2147/CIA.S144651). Also, by using printed art you have one frame shot of a moment the painter choose to zoom on leaving the “looking tense” to a subjective single point interpretation of the painter. How can you assess the degree/intensity of pain relying on that?
We absolutely agree with the fact that in non-verbal cognitive impaired persons, the assessment of pain should comprise the facial, vocalization and body movement items of such assessment scales. In the Discussion, we argue that it is not possible to score items from the category ‘body movements’ in print art due to the fact that print art is motionless (i.e., static). A second limitation of our study is that artists may have felt free to modify the actual expressions that they observed while producing the artwork. Unfortunately, we don’t know how the omission of the categories vocalization and body movement affects the validity of assessing the intensity of pain.